# Characterizing Circulating Tumor Cells and Tumor-Derived Extracellular Vesicles in Metastatic Castration-Naive and Castration-Resistant Prostate Cancer Patients

**DOI:** 10.3390/cancers14184404

**Published:** 2022-09-10

**Authors:** Khrystany T. Isebia, Eshwari Dathathri, Noortje Verschoor, Afroditi Nanou, Anouk C. De Jong, Frank A. W. Coumans, Leon W. M. M. Terstappen, Jaco Kraan, John W. M. Martens, Ruchi Bansal, Martijn P. Lolkema

**Affiliations:** 1Department of Medical Oncology, Erasmus MC Cancer Institute, University Medical Center Rotterdam, 3015 GD Rotterdam, The Netherlands; 2Department of Medical Cell Biophysics, Technical Medical Center, Faculty of Science and Technology, University of Twente, 7522 NB Enschede, The Netherlands

**Keywords:** circulating tumor cells (CTCs), tumor-derived extracellular vesicles (tdEVs), CTCs subclasses, CellSearch, ACCEPT

## Abstract

**Simple Summary:**

The composition of circulating tumor cells (CTCs) and tumor-derived extracellular vesicles (tdEVs) in the blood of 104 patients with metastatic castration-naive prostate cancer and 66 patients with metastatic castration-resistant prostate cancer was compared to explore the potential differences between these biomarkers in the two disease stages.

**Abstract:**

Circulating tumor cell (CTC)- and/or tumor-derived extracellular vesicle (tdEV) loads in the blood of metastatic castration-resistant prostate cancer (CRPC) patients are associated with worse overall survival and can be used as predictive markers of treatment response. In this study, we investigated the quantity/quality of CTCs and tdEVs in metastatic castration-naive prostate cancer (CNPC) and CRPC patients, and whether androgen deprivation therapy (ADT) affects CTCs and tdEVs. We included 104 CNPC patients before ADT initiation and 66 CRPC patients. Blood samples from 31/104 CNPC patients were obtained 6 months after ADT. CTCs and tdEVs were identified using ACCEPT software. Based on the morphology, CTCs of metastatic CNPC and CRPC patients were subdivided by manual reviewing into six subclasses. The numbers of CTCs and tdEVs were correlated in both CNPC and CRPC patients, and both CTCs (*p* = 0.013) and tdEVs (*p* = 0.005) were significantly lower in CNPC compared to CRPC patients. Qualitative differences in CTCs were observed: CTC clusters (*p* = 0.006) and heterogeneously CK expressing CTCs (*p* = 0.041) were significantly lower in CNPC patients. CTC/tdEV numbers declined 6 months after ADT. Our study showed that next to CTC-load, qualitative CTC analysis and tdEV-load may be useful in CNPC patients.

## 1. Introduction 

For metastatic prostate cancer patients, androgen-deprivation therapy (ADT) is used as a first-line treatment. However, resistance to ADT is inevitable, and metastatic castration-naive prostate cancer patients (CNPC) invariably become metastatic castration-resistant prostate cancer patients (CRPC) [1,2]. Metastatic prostate cancer is an extremely heterogeneous disease, and less invasive methods are often sought to gather information about tumor characteristics of metastatic disease and determine the prognosis of these patients [3]. 

Circulating Tumor Cells (CTCs) detected by the CellSearch system reflect disease status and provide a safe and non-invasive biomarker to obtain predictive and prognostic information in various metastatic cancers including CRPC [4,5,6]. Besides CTC counts, the number of tumor-derived extracellular vesicles (tdEVs) in CRPC and other metastatic tumors [7] has also been associated with poor outcomes, and monitoring their frequency may be used to guide therapy.

We have previously reported the application of an open-source, image analysis software, designated ACCEPT (Automated CTC Classification, Enumeration, and PhenoTyping) to evaluate datasets of archived images from prior studies of CTCs captured and analyzed on the CellSearch^®^ platform [8]. In that study and subsequent studies [7,9,10], we observed that elevated levels of circulating large tdEVs, defined here as particles of a size between 1 and 12 μm that co-express epithelial cell adhesion molecule (EpCAM) and cytokeratin (CK) but lack leukocyte-specific CD45 and nuclear staining, were detected in nearly 75% of patients with different metastatic carcinomas. In CRPC patients, tdEVs are present at around 20 times higher numbers than CTCs and have equivalent prognostic power [7]. Whether tdEVs are also associated with overall survival in metastatic CNPC patients is currently being investigated in the Probing intercellular heterogeneity in circulating tumor cells of de novo metastatic CNPC patients (PICTURES) study [11,12]. Next to using CTCs as biomarkers, little is known about the morphological features of CTCs in relation to the different disease stages. Park et al. showed that there were morphological differences between cultured prostate cancer cells and CTCs from CRPC patients [13].

Clearly, CTCs and tdEVs have proven to be of clinical interest in metastatic prostate cancer, since they are associated with overall survival (OS) and could also serve as treatment response markers [6,7,10,14,15,16,17]. However, comparisons of quantity and quality of CTCs, and tdEV abundance, between CNPC and CRPC patients are still lacking. In this study, we compared the number of CTCs, subclasses of CTCs, and tdEVs between CNPC and CRPC patients, and explored the difference between counts at baseline and follow-up, providing insight into the evolution of these biomarkers in different disease stages. 

## 2. Materials and Methods

### 2.1. Patient Inclusion

In this study, we included 104 CNPC patients before ADT initiation and 66 CRPC patients. A total of 94 patients included in this study were enrolled in the CIRCLE study of which 66 were with CRPC and 28 with CNPC [18]. A total of 76 CNPC patients from the PICTURES study were added. A total of 31 of the 76 CNPC patients were followed up after six months of ADT treatment. PICTURES [11,12] is an ongoing study in which CNPC patients with ≥3 CTC per 7.5 mL of blood as assessed by standard CellSearch analysis, undergo Diagnostic Leukapheresis (DLA) to harbor sufficient CTCs for an extensive CTC characterization. In the CIRCLE study [18], patients with metastatic prostate cancer (mPCa) were included and screened for the presence of ≥5 CTCs per 7.5 mL of blood that further underwent a DLA procedure. Biochemical markers analyses and prostate cancer staging was performed according to the standard procedures [11,12,18]. For the CIRCLE study, all patients were histologically or cytologically confirmed for carcinoma of the prostate and measurable metastatic lesion(s) were present according to Prostate Cancer Working Group 2 (PCWG2) and/or Response Evaluation Criteria in Solid Tumors (RECIST 1.1) criteria (i.e., M1 classification according to the 2009 TNM classification) [18,19,20]. CRPC patients were identified by the treating physician as patients with clinical progression (PSA progression according to PCWG2 criteria, radiographical assessment, or both according to PCWG2 criteria). The CRPC patients are a heterogenous group with various numbers of upfront treatments [18]. For the PICTURES study, all patients with metastatic lesions were included according to PCWG3 and/or RECIST criteria [11,12]. The imaging techniques used were not specified but were done as per the treating physician’s choice. We did not specifically capture these data and therefore cannot quantify the extent of the disease. However, patient selection for the CIRCLE and PICTURES studies mandated patients to have at least 3 CTCs/7.5 mL of blood, to undergo DLA. Hence, the majority of the patients were at the high end of the spectrum [19,20]. All study participants had signed informed consent forms in accordance with the Helsinki declaration and all protocols were approved by the Ethics Committees of the CIRCLE (MEC 16-449) and PICTURES studies (MEC 20-0422). The written informed consent was obtained before any study procedures were performed. 

### 2.2. Sample Preparation and Image Acquisition 

Peripheral blood was drawn into CellSave tubes and processed using the CellSearch^®^ System (Menarini, Silicon Biosystems, Inc., Huntingdon Valley, PA, USA). In this system, CTCs were immunomagnetically enriched with ferrofluids coupled with monoclonal antibodies against epithelial cell adhesion molecules (EpCAM) that are expressed on CTCs of epithelial origin. The enriched cell fraction was then fluorescently labeled with (1) nucleic acid dye 4′,6-diamidino-2-phenylindole (DAPI) to identify the nucleus; (2) monoclonal antibodies against cytokeratin 8, 18, and 19 conjugated with Phycoerythrin (CK-PE) (clone C11 and clone A53-B/A2) to identify cells of epithelial origin; (3) monoclonal antibodies against CD45 conjugated with Allophycocyanin (CD45-APC) (clone HI30), to identify leukocytes as has been previously described [21]. After enrichment and staining, the samples were collected in a cartridge contained in a Magnest^®^ (Menarini, Silicon Biosystems, Inc., Huntingdon Valley, PA, USA) Image acquisition of the cartridge content is performed on the CellTracks Analyzer (CTAII), a semi-automated fluorescence microscope with a computer-controlled X, Y, and Z stages, 12-bit CCD camera, NA 0.45 X 10 objective, mercury arc lamp, and filter cubes for FITC/DiOC, PE, APC, and DAPI. To cover the surface of the complete cartridge, an image data set consisting of four-layer tiff files of 175 images was generated. CTCs are originally enumerated after a manual review of thumbnails containing DAPI+, and CK+ objects presented to the reviewer [9,22]. However, in our study, the ACCEPT software was used to automatically enumerate CTCs and tdEVs (Figure 1).

### 2.3. CTC and tdEV Enumeration with ACCEPT 

The 175 four-layer tiff files from 170 patients were analyzed using the Full Detection processor of the open-source ACCEPT software (http://github.com/LeonieZ/ACCEPT, accessed on 12 September 2019) [8]. With the full detection processor, the signal from each of the four-layer tiff files i.e., DAPI, PE, APC, and FITC was segmented, detecting all fluorescent objects. A total of 10 morphological and fluorescence signal intensity measurements for each object were determined in each fluorescent channel. These measurements were used to design and optimize linear gates to identify different classes of objects and enumerate objects falling within them once applied. For this study, a CTC gate and a tdEV gate were defined [7]. The definitions of the gates used are provided in Table 1. All objects falling in the CTC gate were reviewed and the ones identified as debris or artifacts were excluded from the counts, resulting in the CTC ACCEPT corrected counts.

### 2.4. Subclassification and Scoring of ACCEPT Identified CTCs 

CTC ACCEPT corrected counts from all samples were subjected to an unbiased review by three trained individuals to be classified into six subclasses based on their size and morphology: Class I: Clusters, Class II: Pretty, Class III: Heterogenous CK, Class IV: Cleaved CK, Class V: Fragmented DNA/CK, and Class VI: Cleaved CK + fragmented DNA. To allow the individuals to classify the images, a custom scoring tool was developed in MATLAB 2019b, similar to the scoring tool developed for the subclassification of CTCs [8]. The tool was also used to identify the discrepancies between the individuals and obtain a consensus on the classification performed. A full overview of the study is provided in Figure 1.

### 2.5. Statistical Analysis 

The statistical analysis was performed using SPSS 23.0 (SPSS Inc., Chicago, IL, USA). The baseline characteristics were tested with a *t*-test for the normally distributed characteristics, and otherwise with the chi-square or Mann–Whitney U test. The two-tailed Spearman Rho test was performed to evaluate the relationship between the CTCs and tdEVs in CNPC and CRPC patients, respectively. The Mann–Whitney U test was performed to compare the equality of distributions of CTCs and tdEVs between CNPC and CRPC patient cohorts and also of the CTC subclasses. A paired *t*-test was performed to compare the CTC and tdEV counts at baseline and post ADT therapy in CNPC patients.

## 3. Results 

### 3.1. Patient Demographics and Screening

In this study, a total of 104 CNPC patients and 66 CRPC patients were screened for the presence of CTCs. The patient demographics of CNPC and CRPC patients are provided in Table 2. Significant differences were observed in the baseline characteristics of the two patient populations: WHO PS (*p* ≤ 0.001), Hemoglobin (*p* = 0.032), and M-stage at initial diagnosis (*p* ≤ 0.001). The CNPC patients included in the PICTURES study were all metastasized at diagnosis, which is due to the inclusion criteria of this study. 

### 3.2. CTC and tdEV Enumeration Using CellSearch and ACCEPT 

The CellSearch/CTAII images were further analyzed with the Full Detection processor of the ACCEPT software, and the CTC and tdEV gates were applied to yield the CTC ACCEPT and tdEV ACCEPT counts, respectively. The CTC ACCEPT counts were further reviewed by three individuals to eliminate any non-CTCs/artifacts detected, and are called CTC ACCEPT corrected. 

The CTC (CTC ACCEPT corrected) and tdEV counts (tdEVs ACCEPT) from CNPC and CRPC patients were firstly analyzed to assess differences in the disease stages. Spearman Rho correlation of the CTCs (CTC ACCEPT corrected) and tdEVs (tdEVs ACCEPT) indicated strong correlations for CNPC (*p =* 0.84) and CRPC (*p* = 0.92) (Appendix A). This correlation between CTCs and tdEVs was not affected by the disease stage as seen in Figure 2A. The cohorts based on their CTC numbers (CTC = 0, 1 ≤ CTC < 5 and CTC ≥ 5), show comparable tdEV distribution between the CNPC and CRPC patients CTC = 0 (*p* = 0.937), 1 ≤ CTC < 5 (*p* = 0.3) and CTC ≥ 5 (*p* = 0.806) (Appendix A). However, tdEVs may be a more sensitive biomarker as they are detectable even in patients in both cohorts with 0 CTCs (Figure 2B). 

The CTC and tdEV distributions were further analyzed in CNPC and CRPC patients. The CTC (CTC ACCEPT corrected) distribution in CNPC ranged from 0–851 (median 1, SD 102.72, mean rank = 78.28) and was significantly different with *p* = 0.013 (Mann–Whitney U Test, Appendix A) from the CTC distribution observed in CRPC which ranged from 0–229 (median 3.5, SD 38.60, mean rank = 96.88). The frequency of cases having CTCs is lower in CNPC than in CRPC (Figure 3). This can be attributed to the presence of a high frequency of 0 CTC patients in the CNPC cohort. A chi-square test showed that having CTCs (CNPC n = 56, CRPC n = 48) or not (CNPC n = 48, CRPC n = 18) was different between CNPC and CRPC patients (*p* = 0.014).

In case of the tdEVs, their distribution in CNPC ranged from 0–3373 (median 11, SD 558.51) and was significantly different (*p* = 0.005, CNPC mean rank = 77.05, CRPC mean rank = 98.81 Mann–Whitney U Test) from the tdEV counts observed in CRPC, which ranged from 0–2196 (median 49.5, SD 426.07). The number of tdEVs was an order of magnitude larger as compared to CTC counts, in both CNPC and CRPC patients (Figure 2). We also observed that the distribution of tdEVs shifts as the disease progresses from CNPC to CRPC (Figure 4A,B). The chi-square test revealed a significance of *p* = 0.004 when comparing the groups of tdEVs ≤ 50 and tdEVs > 50 between the disease stages, showing that CNPC patients were often in the group tdEVs ≤ 50.

### 3.3. CTC Subclassification 

To investigate whether a difference exists in the composition of the CTCs of CNPC and CRPC, all CTC ACCEPT corrected images were reviewed manually by three trained individuals and sorted into the six different subclasses based on distinct morphological features.

Representative examples of thumbnails assigned in each of the six CTC subclasses are shown in Figure 5. For each of the six CTC subclasses, four examples are provided ranging from CTCs that barely made it into the classification (presented on the left) and increasing likelihood of belonging to the class (presented on the right). Class 1: CTC clusters, defined as two or more nuclei within the CK staining; Class 2: Pretty CTC, defined by intact nuclei with homogeneous and a strong presence of CK within the contour of the CTC; Class 3: Heterogeneous CK, includes CTCs with non-uniform CK staining distribution, irregular shape of CK and nucleus, or CK with protruding nuclei; Class 4: Cleaved CK, indicates the CTCs that are undergoing apoptosis characterized by round speckles of CK [23,24]; Class 5: Fragmented DNA/CTC, indicates CTCs with a fragmented nucleus or CK; Class 6: Cleaved CK + Fragmented DNA, contains apoptotic CTCs with speckled CK and broken nuclei. Objects not correctly classified by ACCEPT according to the CD45-, CK+ DAPI+ cell criteria of CTCs were excluded from the analysis [21].

### 3.4. Distribution of CTC Subclasses in CNPC and CRPC 

In both patient cohorts, we observed that the overall distribution of CTCs into morphologically different CTC subclasses is very similar between the two disease stages (Figure 6A). The percentage of patients (CNPC and CRPC) showing the presence of each CTC subclass is shown in Figure 6B. We observed higher percentages of CRPC patients exhibiting every CTC subclass compared to CNPC patients (Figure 6B). 

When comparing the distribution of the CTC subclasses, we observe that CTC clusters (*p* = 0.006, CNPC mean rank = 79.27, CRPC mean rank = 95.31, Mann–Whitney U Test) and heterogenous CK CTCs (*p* = 0.041, CNPC mean rank = 79.69, CRPC mean rank = 94.66, Mann–Whitney U Test) are significantly higher in CRPC as compared to CNPC patients (*p* < 0.05) (Figure 7). Despite indicating a similar distribution, the variation (SD) observed in all CTC subclasses was higher in CNPC than in CRPC patients (Appendix A and Figure 7). Interestingly, CTCs with heterogeneous CK was the most abundant subclass in CNPC and CRPC patients.

tdEVs numbers were also correlated with the CTC counts of each subclass in both cohorts. The strongest correlation observed in CNPC patients was between the tdEVs and the heterogenous CK CTC subclass (Spearman Rho coefficients ρ = 0.81). In CRPC patients, the correlation was strongest between tdEVs and heterogenous CK CTCs (Spearman Rho coefficients ρ = 0.86) and, between tdEVs and cleaved CK CTCs (Spearman Rho coefficients ρ = 0.82).

### 3.5. Presence of CTC and tdEV Pre- and Post-Therapy in CNPC

We explored the CTC counts and tdEV counts for the CNPC patients included in the PICTURES study at baseline and after 6 months of treatment with ADT. We included 31 CNPC patients for whom the counts were available at both time points. In this analysis, as can be seen in Figure 8, almost all patients showed a decrease of statistical significance in CTC (*p* = 0.012) and tdEV (*p* = 0.042) numbers at 6 months after ADT compared to the baseline.

## 4. Discussion

The clinical validity for CTCs and tdEVs is well established in CRPC, as they were associated with overall survival (OS) and could serve as treatment response markers [6,7,10,14,15,16,17]. However, the comparison of quantity and quality of CTCs and tdEV abundance between CNPC and CRPC is still lacking. In this study, we compared the number of CTCs, subclasses of CTCs, and tdEVs between CNPC and CRPC patients, and explored the difference between counts at baseline and follow-up, providing insight into the evolution of these biomarkers in different disease stages.

Our analysis using the ACCEPT software showed that CTCs can be detected in both disease stages. When comparing the CTC counts between the CNPC and CRPC settings, we conclude that CTC numbers are higher in the CRPC setting. Furthermore, the chance of having no CTCs in CNPC patients is higher than in CRPC. The lower number of CTCs in CNPC could be clinically relevant because the well-known cut-off for prognostication based on CTC counts which are currently set at 5 cells per 7.5 mL of blood may be less suitable as it may underestimate patients at high risk for early progression in the CNPC setting. Importantly, we observed a strong correlation between tdEVs and CTCs irrespective of the disease stage. Furthermore, tdEVs are present in higher numbers than CTCs, and only a few patients have undetectable tdEVs. Moreover, in CNPC patients with no CTCs, tdEVs are often detected. Since the tdEVs are present in higher numbers than CTCs, they may better reflect the phenotypic heterogeneity between patients, particularly in the CNPC setting. Thus, determining tdEVs increases the ability to detect events and might improve prognostication of CNPC compared to using CTC counts since studies in CRPC showed that tdEV counts could serve as prognostic biomarkers [7]. Additionally, in CNPC patients tdEV numbers dropped under the therapeutic pressure of ADT, and this drop may also be used to predict treatment outcomes. Therefore, we conclude that counting tdEVs using the ACCEPT software may improve the stratification of CNPC patients and may even serve as a better way to monitor treatment response, especially when CTCs are low.

Besides the numbers of CTCs and tdEVs in CNPC and CRPC patients, we also aimed to investigate whether the quality of CTCs represented in the form of CTC subsets are differentially distributed in CNPC compared to CRPC. We noted that the number of CTC clusters is significantly higher in CRPC even if there is a wider range in the numbers of CTC clusters in the case of CNPC patients. Studies highlighted that CTC clusters have a higher metastatic potential compared to single CTCs, and that CTC clusters have enhanced survival in the bloodstream [25,26]. Since we observed a higher number of clusters in CRPC patients, we could relate this to a more advanced disease stage caused by escape from therapeutic pressure of ADT, for example. In addition, CRPC patients indicate significantly higher CTC levels with heterogeneous CK expression compared to CNPC patients. TdEVs of CRPC patients showed a strong correlation with the CTC subclass of heterogenous CK and one of the CTC apoptotic subclasses of cleaved CK. The fact that tdEVs of CNPC patients have the strongest correlation with the CTC subclass of heterogeneous CK instead of the apoptotic CTC subclasses, namely the CTCs with cleaved CK or/and fragmented DNA suggests that tdEVs are not only a by-product of CTC apoptosis and degradation. That is further supported by the fact that ADT treatment resulted in the decrease of both CTCs and tdEVs. Even if we cannot exclude the possibility that a portion of the tdEVs in our study may be tumor- or CTC-derived apoptotic bodies, our findings rather suggest a similar (active) release of CTCs and tdEVs from the primary tumor or/and the metastatic lesions.

Our effort to further characterize the CTC and tdEV counts in CNPC is becoming especially relevant with the increasing treatment intensification in this setting. Recent studies suggest that CNPC patients benefit from triple treatment using ADT, chemotherapy, and long-term AR targeted therapy using abiraterone, enzalutamide, apalutamide, or darolutamide [27,28,29,30,31,32]. Obviously, the absolute benefit from this more intensive treatment will depend upon the prognostic classification of these patients. Thus, identifying biomarkers that may aid in risk-adjusted treatment stratification is of utmost importance to improve the cost–benefit ratios of these approaches. Our finding is that tdEVs as detected by the FDA-approved CellSearch assay could be especially relevant in making more informed choices. We suggest that our current study may be used as a starting point for larger-scale collection studies of prospective biomarkers such as tdEVs, to improve the stratification of CNPC patients.

In a small subset of CNPC patients, CTC counts were available after 6 months of treatment. We were interested in CTC numbers after 6 months of ADT. We can conclude that treatment with ADT seems to be very potent since all patients have a decrease in CTC and tdEV counts.

One of the limitations of our study could lie in the way of classifying the CTCs, as this was done manually by three individuals. We tried to limit the differences by reaching a consensus on the CTC classification. In case consensus was not reached, a panel of experts reviewed the CTCs. However, there is always a degree of subjectivity when automation is lacking. Furthermore, the CellSearch isolates only the large tdEVs since it processes plasma-depleted blood samples. Therefore, the herein reported tdEV population probably represents just a small fraction of the total EpCAM+ CK+ DAPI- CD45- tdEVs present in the circulation [7]. Therefore, our data should not be directly compared to studies that use a different extracellular vesicle definition. In future studies, we would like to further characterize the tdEVs to confirm that they are of tumor origin by evaluating the expression of prostate-specific proteins such as PSMA on tdEVs, and performing downstream analyses such as RNA sequencing to confirm the presence of prostate cancer-specific RNAs. Another disadvantage is that CTC counts after 6 months of treatment were only possible in a small subset of patients, since the PICTURES study is an ongoing project.

## 5. Conclusions

In conclusion, we have demonstrated that CRPC patients have higher CTC and tdEV counts than CNPC patients. Qualitative differences in CTCs were observed; CTC clusters and heterogeneously CK expressing CTC were more prevalent in CRPC patients. The ACCEPT software helps us to identify tdEVs, and they may serve as a better way of determining prognosis in CNPC patients. Whether CTC subclasses change under therapeutic pressure of ADT, and whether CTC subclasses and tdEV count could serve as prognostic biomarkers, need to be further elucidated.

## Figures and Tables

**Figure 1 cancers-14-04404-f001:**
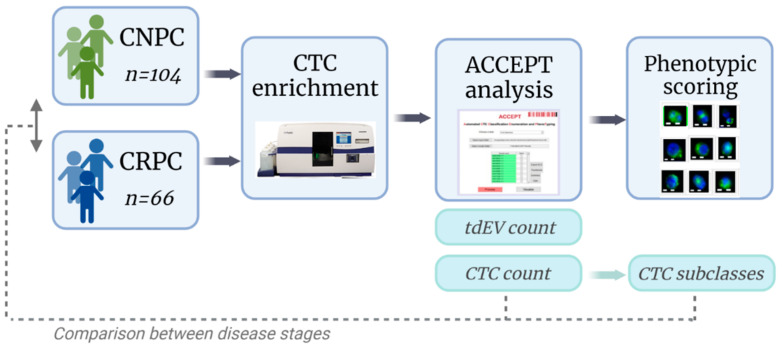
Overview of patient inclusion, CTC enrichment, CTC and tdEV enumeration, phenotypic scoring, and classification of CTCs. Made with Biorender.com.

**Figure 2 cancers-14-04404-f002:**
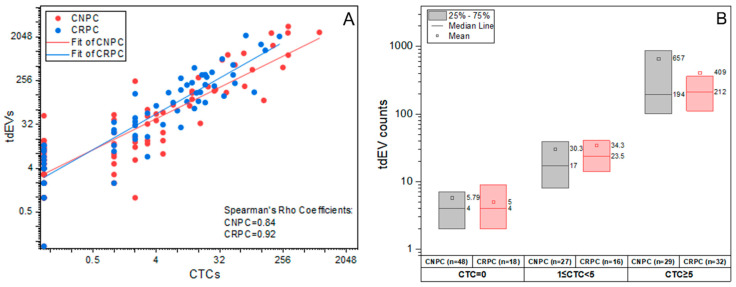
Correlation of CTCs (CTC ACCEPT corrected) and corresponding tdEVs from CNPC (n = 104) and CRPC (n = 66) patients, prior to initiation of therapy. (**A**): CTCs and tdEVs were strongly correlated (Spearman Rho coefficients ρ = 0.84 in CNPC, ρ = 0.92 in CRPC) and this correlation was independent of the disease stage. (**B**): tdEVs within the categories CTC = 0 (CNPC, CRPC median = 4, *p* = 0.937), 1 ≤ CTC < 5 (median CNPC = 17, CRPC = 23.5, *p* = 0.3) and CTC ≥ 5 (median CNPC = 194, CRPC = 212, *p* = 0.806) showed no significant difference between CNPC and CRPC.

**Figure 3 cancers-14-04404-f003:**
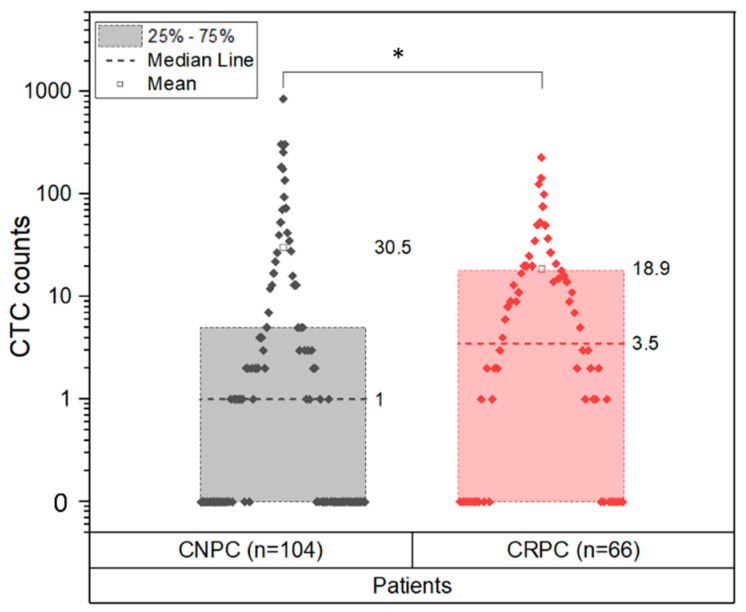
Presence of CTC in CNPC before initiation of therapy and CRPC. The CTC counts obtained from CNPC and CRPC patients were assessed for differences between the patient cohorts. The CTC (CTC ACCEPT corrected) counts of 104 CNPC patients (mean rank = 78.28, median = 1) showed significant difference (*p* = 0.013) compared to the 66 CRPC patients (mean rank = 96.88, median = 3.5). The chi-square test showed that having CTCs (CNPC n = 56, CRPC n = 48) or not (CNPC n = 48, CRPC n = 18) was different between CNPC and CRPC patients (*p* = 0.014). * Significance *p* < 0.05.

**Figure 4 cancers-14-04404-f004:**
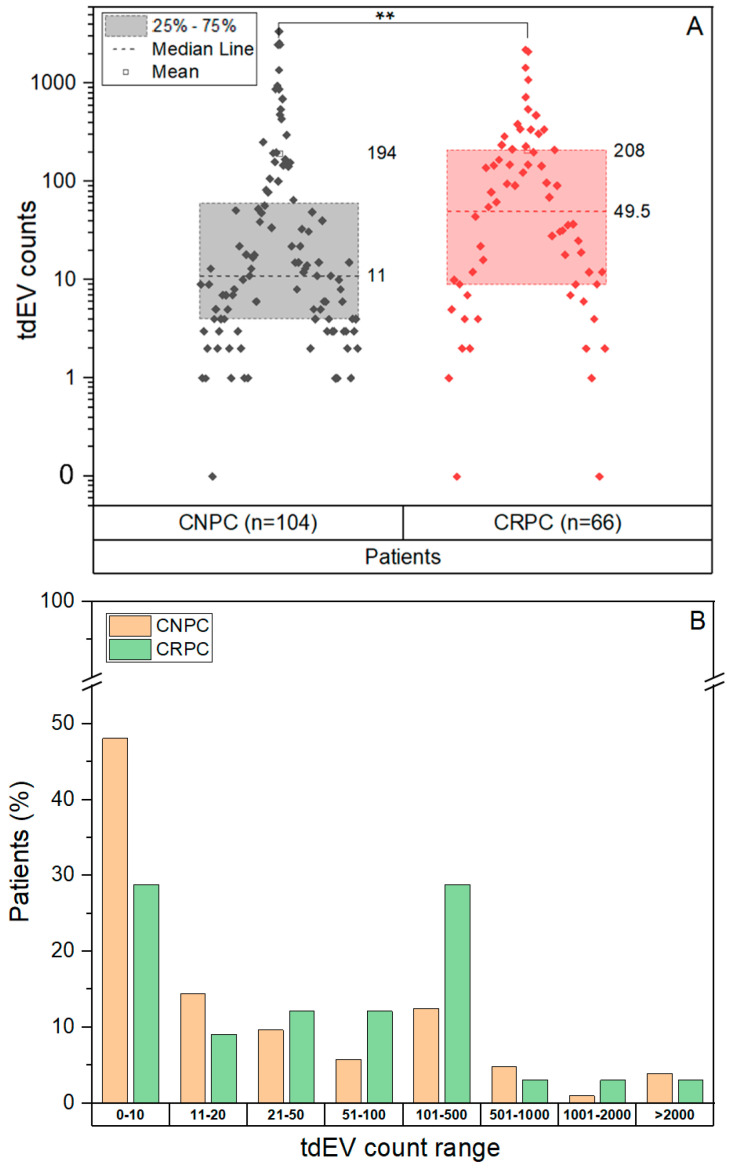
Presence of tdEVs in CNPC and CRPC before initiation of therapy. (**A**): The tdEV counts obtained from CNPC and CRPC patients were tested to check for differences between the patient cohorts. tdEVs showed a significant difference (*p* = 0.005) between CNPC (mean rank = 77.05, median = 11) and CRPC (mean rank = 98.81, median = 49.5). (**B**): The percentage of patients sorted on increasing tdEVs in CNPC and CRPC. ** Significance *p* < 0.01.

**Figure 5 cancers-14-04404-f005:**
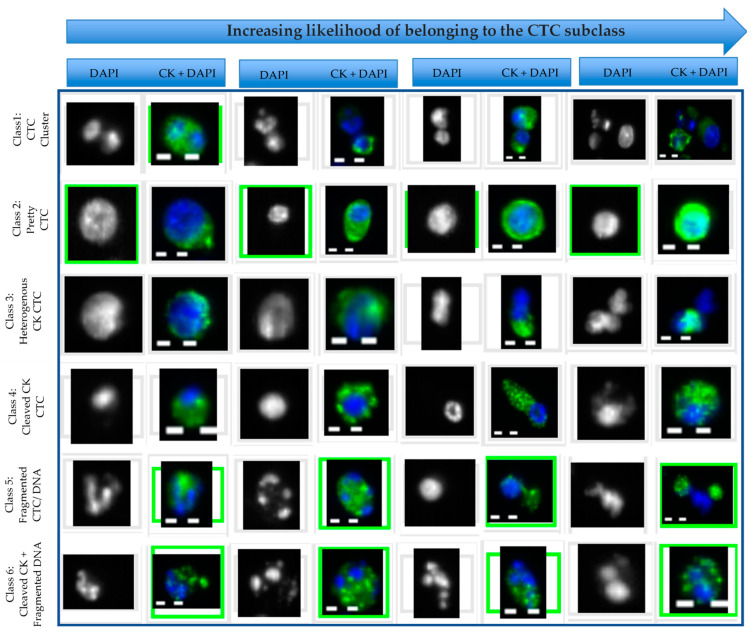
Examples of CTCs assigned to the different subclasses. Based on morphological appearance, six subclasses were identified: Class 1: CTC clusters; Class 2: Pretty CTC; Class 3: Heterogeneous CK CTC; Class 4: Cleaved CK CTC; Class 5: Fragmented DNA/CTC; Class 6: CTC with Cleaved CK + Fragmented DNA. For each class, the objects are shown with an increasing likelihood of belonging to the class. Composite images are shown with the nuclear stain DAPI colored blue and the cytokeratin (CK) staining colored green.

**Figure 6 cancers-14-04404-f006:**
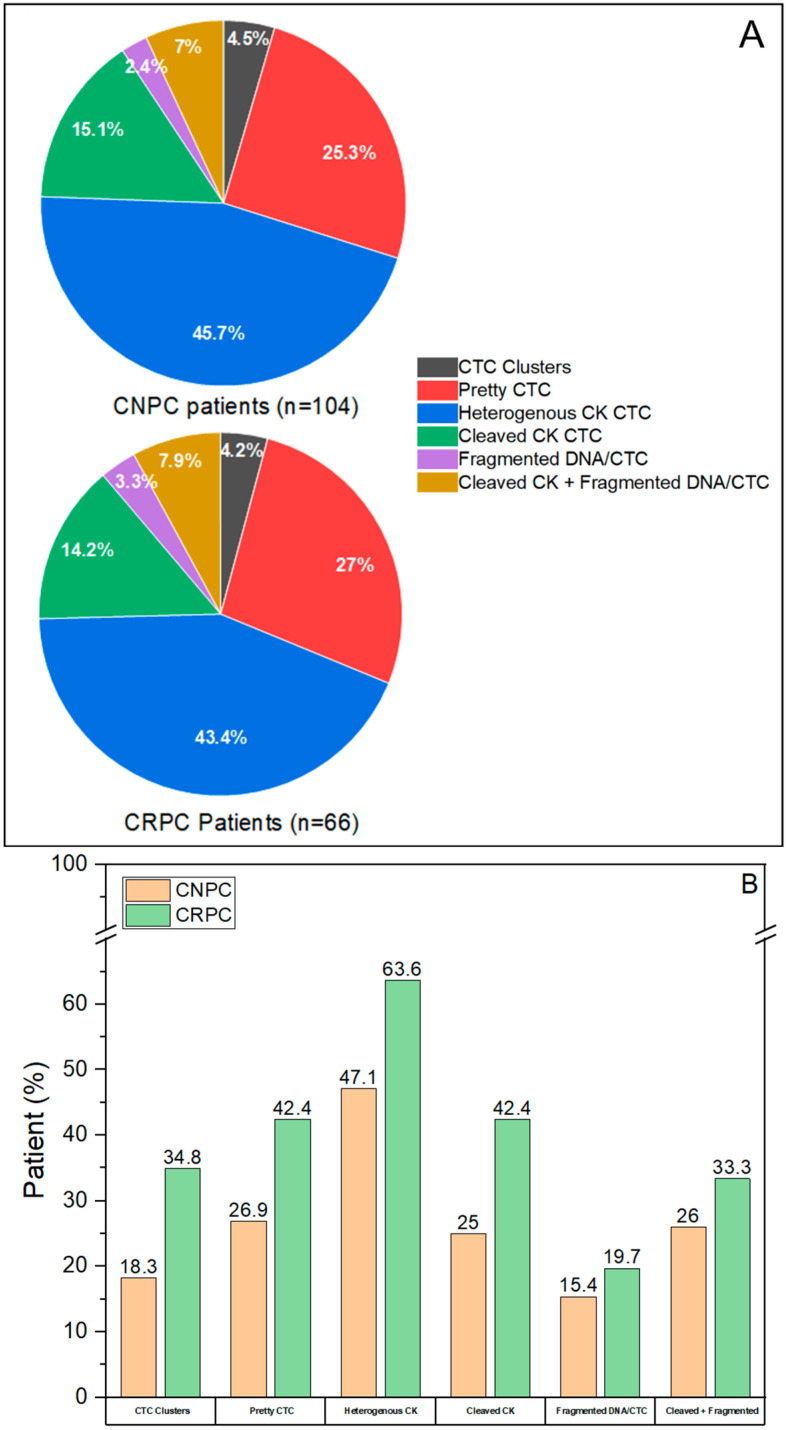
CNPC (n = 104) and CRPC (n = 66) patients exhibiting CTC subclasses. (**A**): Distribution (%) of total CTCs of CNPC and CRPC patient cohorts into the different CTC subclasses (**B**): Patients (%) in CNPC and CRPC showing the presence of CTC subclasses. A higher percentage of CRPC patients have CTCs in each CTC subclass.

**Figure 7 cancers-14-04404-f007:**
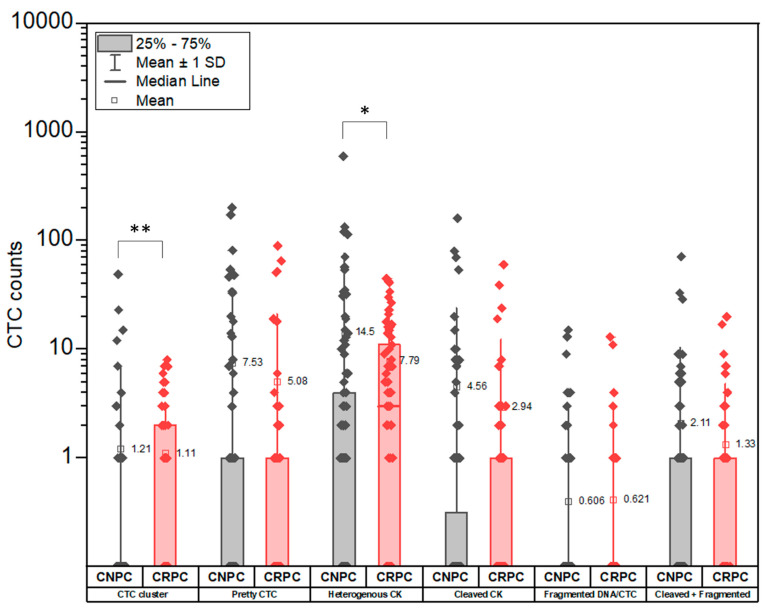
Comparison of CTC subclass distribution in CNPC and CRPC patients. The CTC counts in CTC clusters (*p* = 0.006, CNPC mean rank = 79.27, CRPC mean rank = 95.31, Mann–Whitney U Test) and Heterogeneous CK classes (*p* = 0.041, CNPC mean rank = 79.69, CRPC mean rank = 94.66, Mann–Whitney U Test) showed significant differences between the CNPC and CRPC patients. CTC classes in CNPC showed higher SD compared to CRPC. * Significance *p* < 0.05 ** Significance *p* < 0.01.

**Figure 8 cancers-14-04404-f008:**
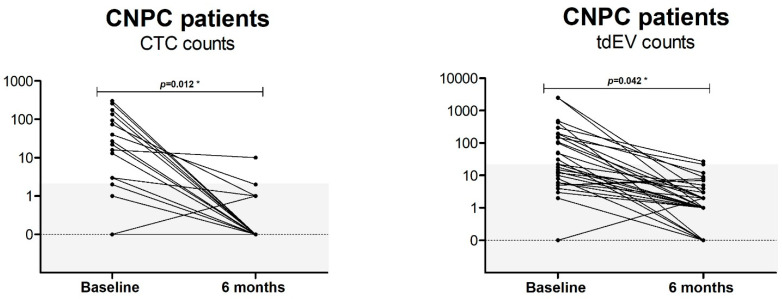
Baseline CTC/tdEV counts and CTC/tdEV counts after 6 months of ADT of 31 CNPC patients. The gray area indicates the background CTC and tdEV count detected in healthy individuals. The decrease in CTC and tdEV counts pre- and post-therapy were statistically significant with *p* = 0.012 for CTCs and *p* = 0.042 for tdEVs. * Significance *p* < 0.05.

**Table 1 cancers-14-04404-t001:** ACCEPT gates for CTC and tdEV identification.

Parameters	CTCs	tdEVs
Mean intensity CD45	≤5 AU	≤5 AU
Mean intensity DAPI	>45 AU	≤5 AU
Mean intensity CK	>60 AU	>60 AU
max intensity CK	-	>90 AU
Area CK	-	≤150 μm^2^
Eccentricity CK	-	≤0.8 AU
Perimeter to area CK	-	≤1 AU
Perimeter CK	-	>5 pixels
CK size	>16 μm^2^	-
CK size	≤400 μm^2^	≤150 μm^2^
CK overlay with DAPI	>0.2 AU	-
Mean intensity marker 1 (PE)	≤5 AU	≤5 AU
Mean intensity marker 2 (FITC)	≤5 AU	≤5 AU

AU = Arbitrary Units.

**Table 2 cancers-14-04404-t002:** Baseline characteristics.

	CNPC	CRPC	
Characteristic	(N = 104)	(N = 66)	
**Age at registration**			*p* = 0.499
Median (range)—years	70 (49–83)	69 (49–82)	
**WHO PS at registration—no. (%)**			*p* ≤ 0.001 ***
- 0	65 (62.5%)	16 (24.2%)	
- 1	35(33.7%)	50 (75.8%)	
- 2	4 (3.8%)		
**Initial PSA at primary diagnosis, µg/L**			*p* = 0.176
Mean ± SD	647.3 ± 1665.8	334.5 ± 900.4	
Median (range)	107 (3.0–11,098.0)	38 (3.0–4786.0)	
**Hemoglobin—g/L**			*p* = 0.032 *
Mean ± SD	8.4 ± 1.3	7.9 ± 1.0	
Median (range)	8.5 (4.3–10.6)	8 (5.5–9.7)	
**Alkaline phosphatase—IU/L**			*p* = 0.192
Mean ± SD	392.7 ± 1157.2	156.8 ± 144.1	
Median (range)	116(45.0–8854.0)	104 (58.0–894.0)	
**Lactate dehydrogenase—IU/L**			*p* = 0.203
Mean ± SD	243.6 ± 89.6	313.3 ± 408.4	
Median (range)	223.5 (139.0–644.0)	202.5 (153.0–2718.0)	
**Albumin—g/L**			*p* = 0.156
Mean ± SD	41.6 ± 5	43.1 ± 3.5	
Median (range)	42.5 (27.0–51.0)	43 (35.0–54.0)	
**Gleason score by diagnosis**			*p* = 0.165
Median (range)	8 (6.0–10.0)	9 (5.0–10.0)	
**M-stage at diagnosis**			*p* ≤ 0.001 ***
- M0	5 (4.8%)	20 (30.3%)	
- M1a, M1b, M1c	88 (91.4%)	30 (50%)	
- Mx	4 (3.8%)	13(19.7%)	
**Type of prior therapy**			
Docetaxel		35	
Cabazitaxel		8	
Enzalutamide		27	
Abiraterone		6	
Other systemic therapy		6	

Range = min.–max., WHO PS = World Health Organization Performance Score, SD = Standard Deviation, PSA = prostate-specific antigen. CNPC patients missing information: iPSA (n = 3), Hemoglobin (n = 14), Alkaline phosphatase (n = 43), Lactate dehydrogenase (n = 44), Albumin (n = 56), Gleason score (n = 10), M-stage at diagnosis (n = 7). CRPC patients missing information iPSA (n = 4), Hemoglobin (n = 9), Alkaline phosphatase (n = 24), Lactate dehydrogenase (n = 24), Albumin (n = 35), Gleason score (n = 6), M-stage at diagnosis (n = 3). Chi-square test for the M-stage and WHO status. Mann–Whitney U test for Gleason score. *t*-test for age, iPSA, Hemoglobin, Alkaline Phosphatase, Lactate dehydrogenase, and Albumin. * Significance *p* < 0.05; *** Significance *p* < 0.001.

## Data Availability

The data presented in this study are available on request from the corresponding author. The data is not publicly available as the PICTURES is an ongoing study.

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
