# Peer review of "Characterizing Circulating Tumor Cells and Tumor-Derived Extracellular Vesicles in Metastatic Castration-Naive and Castration-Resistant Prostate Cancer Patients"

_cancers, 2022, doi:10.3390/cancers14184404_

Round 1

Reviewer 1 Report

Congratulations to the authors. Very interesting study that may have an impact on therapies against prostate cancer.

Minor revision:

1.-Table 2 (results) shows biochemical and hematological parameters and staging criteria for prostate cancer. Therefore, some description of them should be included in the method.

As a suggestion,  you can include de folowing paragraph  "biochemical markers and prostate cancer staging was performed according to standard procedures [references]"

Author Response

Journal: Cancers Special Issue “The 5th ACTC: “Liquid Biopsy in Its Best”

Manuscript title: Characterizing circulating tumor cells and tumor-derived extracellular vesicles in metastatic castration-naive and castration-resistant prostate cancer patients

Manuscript ID: cancers-1867067

Authors: Khrystany T Isebia*, Eshwari Dathathri*, Noortje Verschoor, Afroditi Nanou , Anouk C de Jong , Frank AW Coumans , Leon WMM Terstappen , Jaco Kraan , John WM Martens , Ruchi Bansal and Martijn P Lolkema

The authors would like to thank the reviewers for their valuable comments which, we have addressed in the manuscript. Please find the point-to-point responses and explanation of the changes made in the revised manuscript. The changes are tracked in the revised manuscript. We hope that we have addressed all the reviewer’s concerns satisfactorily.

Reviewer 1:

Comment: Table 2 (results) shows biochemical and hematological parameters and staging criteria for prostate cancer. Therefore, some description of them should be included in the method.

As a suggestion,  you can include de following paragraph  "biochemical markers and prostate cancer staging was performed according to standard procedures [references]"

Response: We thank the reviewer for this comment. We have now included the above-suggested paragraph with references in Section 2. Please refer to Materials and Methods, Subsection 2.1 Patient Inclusion (Page 3).

Reviewer 2 Report

In this study, the authors leverage the CellSearch technology to quantify blood levels of circulating tumor cells (CTCs) and tumor-derived extracellular vesicles (tdEVs) in two populations of patients with castration-naive prostate cancer and castration-sensitive prostate cancer.  

1) Patient population: Additional information is required in the Patient characteristics section. All CNPC patients were metastatic at diagnosis. Imaging techniques used for the diagnosis need to be mentioned. Were they all widespread metastatic or some patients presented with oligometastatic disease? What prior treatments CRPC patients received? How CRPC patients were identified? PSA progression, radiographically or both?

2) In Table 2, Median +/- range for each characteristics would be more appropriate. What does initial PSA means? At first diagnosis? Both cohorts have a mean PSA > 300 ug/L which is very high and not representative of patient cohorts found in clinical trials for prostate cancer. These patients are more on the high end of the spectrum. Correlation of CTC levels and tdEVs with PSA would be relevant to include and see how sensitive is the assay in patients with PSA <100 ug/L. 

3) The authors showed a very high correlation of CTC levels and tdEV levels in both CNPC and CRPC cohorts. It makes sense since the authors previously reported that the tdEVs may origin from apoptotic CTCs (Coumans FAW, Annals of Oncology 2010). That being said, it is not clear what the authors want to demonstrate with this result?

4) Is it accurate to use the term tdEVs to define EpCAM+CK+ objects? Did they author demonstrate tdEVs origin from prostate cancer? In Figure 8, the gray area indicates tdEV counts in healthy individuals which suggests a non-cancer origin of some of EpCAM+CK+ objects. Further characterization of these objects is needed to justify the term tdEVs. 

5) In Figure 8, the authors showed that ADT treatment can lead to strong reduction in levels of CTC and tdEVs which is very interesting. Was it statistically significant? This result has limited significance since no association with oncological outcomes is provided. Are the data too early to do such analysis?

6) CTC and tdEVs are emerging prognostic liquid biomarkers in prostate cancer.  To gain more reach, the authors should clarify the translational relevance of their study. To date, it seems very descriptive and it is difficult to capture how such technology can improve the management of patients with prostate cancer. 

Author Response

Dear Reviewer, 

Thank you.

Reviewer 3 Report

Good manuscript on "Characterizing circulating tumor cells and tumor-derived extra-cellular vesicles in metastatic castration-naive and castration-resistant prostate cancer patients" I recommend publication of this manuscript.

Minor comments:

The authors should check the grammatical errors.

Author Response

Journal: Cancers Special Issue “The 5th ACTC: “Liquid Biopsy in Its Best”

Manuscript title: Characterizing circulating tumor cells and tumor-derived extracellular vesicles in metastatic castration-naive and castration-resistant prostate cancer patients

Manuscript ID: cancers-1867067

Authors: Khrystany T Isebia*, Eshwari Dathathri*, Noortje Verschoor, Afroditi Nanou , Anouk C de Jong , Frank AW Coumans , Leon WMM Terstappen , Jaco Kraan , John WM Martens , Ruchi Bansal and Martijn P Lolkema

The authors would like to thank the reviewers for their valuable comments which, we have addressed in the manuscript. Please find the point-to-point responses and explanation of the changes made in the revised manuscript. The changes are tracked in the revised manuscript. We hope that we have addressed all the reviewer’s concerns satisfactorily.

Reviewer 3:

Comment: The authors should check the grammatical errors.

Response: We have proofread the manuscript thoroughly for grammatical errors and corrected them using the Grammarly Software.

Round 2

Reviewer 2 Report

I thank the authors for addressing all my comments.